# Self-Sensing Antenna for Soil Moisture: Beacon Approach

**DOI:** 10.3390/s22249863

**Published:** 2022-12-15

**Authors:** Maja Škiljo, Zoran Blažević, Lea Dujić-Rodić, Toni Perković, Petar Šolić

**Affiliations:** Faculty of Electrical Engineering, Mechanical Engineering and Naval Architecture in Split, University of Split, 21000 Split, Croatia

**Keywords:** antenna design, Internet of things, Internet of underground things, LoRa, PIFA, matching, radio sensing, soil electrical properties, soil irrigation, agriculture

## Abstract

On the way from the Internet of things (IoT) to the Internet of underground things (IoUT) the main challenge is antenna design. The enabling technologies still rely on simple design and low cost, but the systems are more complex. The LoRa-based system combined with a machine learning approach can be used for the estimation of soil moisture by using signal strength data, but for the improvement of the system performance we propose the optimization of the antenna for underground use. The soil properties are frequency-dependent and varying in time, which may cause variations in the signal wavelength and input impedance of the antenna underground. Instead of using wideband antenna design or standard helical antenna provided in LoRa module, which are typical in the IoUT research community for communication links, we propose a narrow-band antenna design for the application in soil moisture sensing. It is shown that the approach of simply matching the antenna buried in dry sand can provide a substantial signal level difference, ranging from approximately 10 dB (achieved by proof-of-concept measurements) to as much as 40 dB (calculated by a full wave simulator) in reflection coefficient when the moisture content is being increased by 20%. This can ensure more reliable radio sensing in novel sensorless technology where soil moisture information is extracted from the signal strength of a transmitting device.

## 1. Introduction

As the development of IoT grows rapidly in above ground (AG) applications, there is also a great expansion of Internet of underground things in underground (UG) and under- to above-ground (UG-AG) applications, e.g., pipeline monitoring, smart farming, and precision agriculture [1,2,3]. Likewise, simple design and cost-effective solutions are one of the focuses of the IoUT systems development in order to bring efficiency together with low cost worldwide [4,5].

In our previous work [6], a cost-effective and low-power LoRa-based soil moisture sensing device is proposed for the estimation of the soil moisture by using the collected signal strength data. By using machine learning (ML) techniques, it is possible to achieve a good accuracy in soil moisture estimation when using this approach. The reliability is the main problem of the IoUT links because the antennas are buried underground where varying and uncertain soil properties influence both the propagation of radio waves and the antenna characteristics [1,7]. As the accuracy and reliability of the system depends greatly on the difference of the received signal strength values received from the antenna buried in dry and irrigated soil, the motivation of this work is to design a new antenna with a matching method in order to increase this signal level difference, and consequently improve the system performance.

A theoretical model for a dipole buried underground for IoUT applications, given in [1], is based on extensive experiments in different soils at different antenna depths and water content in the soil. The antenna properties are shown to be highly dependent of soil properties, moisture content, frequency, and burial depth, as the antenna resonant frequency shifts and quality factor varies when the antenna is placed underground. The model takes into account the direct, reflected, and lateral path of the radio wave, and predicts resonant frequency of the dipole buried underground. Consequently, the proposed antenna is a wideband planar antenna suitable for IoUT communication because it accounts for soil moisture variations by covering a wide range of frequencies [1]. There are other similar wideband antenna designs [3,8,9] that are optimized to work well in IoUT communication links for all soil moisture contents. On the contrary, the research community of IoUT often simply uses the LoRa module with its embedded helical antenna, without adjusting it and optimizing for the use underground [10,11].

In this paper, instead of using a standard helical LoRa antenna, a new antenna design method is considered in order to achieve higher accuracy in soil moisture estimation for soil irrigation applications. When the narrow-band antenna designed for AG communications, is placed underground, its resonant frequency shifts, and its size needs to be reduced due to the reduction of signal wavelength in the soil (caused by higher dielectric constant of the soil) [1,9]. Moreover, the variation of antenna characteristics in the soil is not controlled and usually does not give straightforward results and conclusions regarding the difference between irrigated and dry soil. Consequently, based on our previous work [12], the use of the antenna tuned and matched in dry soil can provide better signal level differences between the irrigated and dry soil, and additionally a higher signal level above ground. We propose the use a short-circuit type of antenna, printed inverted F antenna (PIFA), with antiresonance occurring first, allowing the possibility to tune the antenna by using only the trimmer capacitor in series. Here, we provide the proof-of concept measurements of the proposed antenna in the controlled environment in order to investigate the feasibility of the approach. Based on our previous work [6,12,13], this approach together with the ML techniques, can improve soil moisture estimation by using the signal strength approach in the proposed novel sensorless low-power wide-area network (LPWAN) radio technology.

The paper is organized as follows. The hardware of the LoRa-based system for soil moisture sensing is described in Section 2 along with the sensing algorithm, analysis of the collected data and derived conclusions in Section 3. In Section 4, the motivation and the proposed antenna design for the system improvement are described. Section 4 deals with the practical antenna design and its proof-of-concept measurements in sand with different levels of soil moisture, the results of antenna reflection parameter and received power. Finally, the conclusion with the future work is given in Section 6.

## 2. Hardware Implementation of LoRa-Based Soil Moisture Sensing

LoRa operates on a license-free industrial, scientific, and medical (ISM) frequency which is characterized by low data rates, different communication modes, data transfer speed, installation cost and openness to end users. LoRa as an LPWA technology uses the chirp-spread spectrum modulation (CSS), while LoRaWAN is one of the most widely used LPWA technologies, and implements LoRa radio technology. LoRaWAN represents the medium access control (MAC) protocol, aimed at low-power (primarily battery-operated) devices. The first specification for this protocol was published in 2015 and a new specification was published in 2017. The ability to provide long-range communication depends on the CSS technique and nonlicensed sub-1 GHz transmission frequency range (e.g., 863 to 879 MHz for Europe and 433 MHz for Europe). The parameters required for uplink and downlink communications between LoRa modules and gateways, such as spreading factor (SF) and bandwidth (BW), are defined for each frequency plan. As defined by the LoRaWAN specification, in the EU range, end devices must be able to operate on at least 16 channels within the frequency range of 863 to 870 MHz. The first three channels are determined to correspond to the frequencies of 868.1, 868.3, and 868.5 MHz, guaranteed to be implemented between end devices and gateways.

LoRaWAN uses a star network topology (Figure 1) with three main participants: LoRa modules (end nodes) that can be designed for different applications, a single (or more) LoRa gateway, and a central network server. The gateway forwards the packets received between the LoRa module and the central network server. The central server then relays the packets received to the application server and processes them for further application purposes.

The ML-based soil moisture sensor is based on the Arduino Pro Mini (with ATmega328P), which operates at a 3.5 V input voltage. To enable LoRaWAN-based communications, an RFM95W module with a SX1276 chip and a spring antenna with +14 dB transmission power was used. To save energy during the inactive period, the sensor device uses TPL5110 NanoTimer that cuts off power for a predefined time period (e.g., 5 min), increasing the battery lifetime. The sensor device was buried 14 cm under the ground with antenna vertically oriented, within the range of two LoRaWAN gateways as shown in Figure 2. Once the message has reached the base stations, it is forwarded to the TTN network and application server. (Figure 1). Additionally, TTN allows the sender to forward messages from its infrastructure to our servers by using the MQTT protocol. Figure 3 shows a snapshot of soil moisture with RSSI and SNR values recorded at one gateway. As can be seen, with increased soil moisture, both RSSI and SNR signal values decreased, showing a tight relationship between these values.

## 3. Sensing Algorithm

### 3.1. Analysis of Collected Data from LoRaWAN-Based Soil Moisture Sensor

An exploratory data analysis was performed to discover anomalies in the data, define the required data preparation approach, and identify ML algorithms that might help predict the desired level of soil moisture. The collected sensor data contained information about RSSI, SNR, soil temperature, soil moisture level, timestamp, and LoRaWAN gateway ID. The majority of data was collected within the months of November and December, for a sampling rate of 5 min. In the rest of the text, the RSSI and SNR of Gateway 1 and Gateway 2 will be referred to as RSSI${}_{1}$ and SNR${}_{1}$, and RSSI${}_{2}$ and SNR${}_{2}$, respectively. Key characteristics of raw data variables were tracked by observing changes of signal strength in contrast to soil moisture over time. As can be observed from Figure 3, the change of RSSI is rapid, whereas the soil moisture alters gradually. A trend can be noticed, wherein the increasing moisture affects the signal strength causing its decrease, and vice versa. As a result of channel stochastic behaviour there are two major fading components of the received signal strength. One is the swift variation in signal strength due to the multipath propagation and the other is its slower variant, which is mostly a result of the signal reception in the radio shadow of large obstacles [14].

Hence, the raw data were smoothed by decomposing the received signal strength into long-term and short-term fading factors by using a two-hour time window. The long-term factor component was calculated by taking 24 samples of raw RSSI and SNR data, calculating their mean, and subtracting it from the raw values. Data were then aggregated to assign smoothed RSSI values to soil moisture percentage classes to determine whether RSSI values correlated with specific soil moisture percentage classes.

Calculations of Pearson correlation coefficients [15] between specific classes of soil moisture and the corresponding RSSI and SNR values confirmed previously noticed inverse relationships between signal strength and soil moisture. As can be seen in Figure 4, RSSI${}_{2}$ and SNR${}_{2}$ substantially correlate with soil moisture.

Data obtained from Gateway 1 show that the SNR${}_{1}$ correlates more with the soil moisture than RSSI${}_{1}$, as can be observed from Table 1, indicating that the farther the gateway is, the channel affects RSSI more potent than soil moisture. Data tracing confirmed that there has been a specific working-day timeframe that modified the mean value of the signal strength parameter.

This implies that lower values of SNR and RSSI would indicate higher soil moisture as was observed in Figure 3.

Based on the obtained results, it was concluded that RSSI and soil moisture values were significantly negatively correlated and that a suitable ML algorithm should be able to encompass the complexity of the data properties detailed in the above analysis.

### 3.2. Machine Learning Approach and Model

IoT systems are becoming increasingly dynamic and complex, and ML has been regarded as the key technology for autonomous and smart network management [16]. ML has been considered as one of the most prominent methods for the prediction/estimation based on models and algorithms [17]. These algorithms are capable of finding patterns in massive datasets, particularly in a time series of observational data [15]. The general purpose of ML approaches is to extract information from data (training data) in order to carry out a task, in our case the soil moisture prediction based from obtained data. In recent years deep learning (DL) has been actively employed in IoT applications as one of ML approaches [18]. In contrast to traditional ML algorithms, DL effectively extracts IoT data from noisy and complex environments, and is a powerful analytical tool for big data, yielding improved performance for such jobs [19].

Therefore, the support vector regression (SVR) model and the long short-term memory (LSTM) model were developed, tested, and analyzed as a ML approach to soil moisture prediction. Both models were validated by using the same data and in the same manner, as further specified. Rather than interpretation, the models’ primary goal was the precise calculation of relative soil moisture based on signal strength. All raw RSSI and SNR data samples captured on two LoRaWAN gateways, as well as soil moisture, were used in the models. The models were built in three steps: data preprocessing, model definition, and model validation, as described in the following.

Data normalization was a part of data preprocessing, because of the different value scales of variables in the collected data. In general, relative moisture was as a percentage, whereas RSSI and SNR values have been measured in decibels. The models were fed numerical RSSI and SNR values, and the output was a numeric value that estimated relative soil moisture. Furthermore, for model assessment, data were divided into training and test sets in an 80–20% ratio respectively.

The test set was used to validate the models by using two measures. Namely, the loss functions used for estimation of error were mean squared error (*MSE*) and mean absolute error (*MAE*) defined with Equations (Equation 1) and (Equation 2):(1)$$MSE=\frac{1}{2m}\sum_{i=1}^{m}{\left({\hat{y}}^{\left(i\right)}-y^{\left(i\right)}\right)}^{2}.$$
(2)$$MAE=\frac{1}{m}\sum_{i=1}^{m}\left|{\hat{y}}^{\left(i\right)}-y^{\left(i\right)}\right|.$$

A lower *MSE* indicates greater estimation accuracy. *MSE* calculates the average squared difference between the estimation and the expected results, whereas *MAE* calculates the average magnitude of errors across a set of estimations. Furthermore, validation loss reflects how well or poorly the model performs during training. The computational machine’s specifications include an Intel core i5-7300HQ@2.50 GHz processor, 8 GB of RAM, and an NVIDIA GTX1050 GPU running the 64-bit Windows 10 operating system and the NVIDIA CUDA Deep Neural Network library (cuDNN). The Keras 2.3.1 Python library was used, which was run on top of a Tensorflow 2.2.0.

Detailed model elaboration is presented within the research [6]. The SVR model implemented the RBF kernel. The model’s input was as follows. For each value of RSSI and SNR at time step *t* required for the estimation of moisture at time step *t*, values of RSSI and SNR at time step t−1 were also taken. This provided the model with a “hybrid short-term memory” of previously measured values in time step t−1. The model was verified on the test set, yielding losses of MSE=0.0243 and MAE=0.0487. Figure 5 shows the model’s estimation of soil moisture on the test set compared to expected soil moisture values.

The LSTM model was trained on previously described preprocessed data. In terms of inputs, a time step of 18 was chosen to approximate 90 min of observations (18 samples 5 min period) for each estimation. Normalized data was fed into the LSTM model with the goal of estimating relative soil moisture based on signal strength. Several options for number of neurons per layer, learning rates, epochs, and different optimizers were tested, and the best results were achieved for 32 neurons, on both LSTM layers with a learning rate of 0.001 and the number of epochs being 100. Three optimizers were evaluated: Adam, RMSprop, and SGD. RMSprop outperformed the other two optimizers and was chosen as optimizer for the final model design, achieving MSE and MAE errors of 0.00018 and 0.01043, respectively.

Even with a small and limited dataset with only a few months of representative data (winter period), significant results were obtained for the ML approach for estimating soil moisture from signal strength. Table 2 provides a comparison between the estimation training and test time and the previously described errors. As can be observed from the table, although the LSTM model needs more time to train, it achieves a minimum delay between two consecutive estimations based on the testing time. Furthermore, it has lower MAE and MSE errors in contrast to the SVR model.

To conclude, the SVR model provided a good estimates of soil moisture from RSSI and SNR, revealing that the previously exhibited correlation between SNR, RSSI, and soil moisture was significant. What is more, the SVR model validated the signal strength approach and the premise of moisture sensing by using just the RSSI and SNR data and ML approaches. Finally, by using avaliable data, the stacked LSTM model obtained significantly more accurate estimates of soil moisture and outperformed the traditional SVR in terms of estimation accuracy. This implies that, as a DL model designed for time series data, LSTM was able to better encompass the complex correlation between RSSI, SNR, and soil moisture, resulting in improved performance and precision.

## 4. The Optimization of the IoT Antenna Design for Underground Application

The new results collected during the summer period, shown in Figure 6 provide a different insight into the correlation between RSSI and soil moisture. In the summer period, the soil is mostly dry during a long period and, when it is irrigated (naturally or mechanically), the soil moisture can increase by roughly 15% and RSSI values drop by roughly 6–7 dB on average. This offers a possibility to control the environment for the antenna design and for calibrating the system. The properties of dry soil can be taken from the literature, measured, and the actual soil samples tested in the laboratory (e.g., [20]). The helical LoRa antenna used in this system is tuned and matched at 868 MHz for the use in free space. When placed underground, the antenna characteristics change, especially when the soil is irrigated. This change is not controlled and usually does not give straightforward results and conclusions regarding the difference between irrigated and dry soil. Consequently, the use of the antenna tuned and matched in dry soil can provide better signal level differences between the irrigated and dry soil, and additionally a higher signal level above ground [12]. Consequently, this should lead to even higher precision of the ML approach for estimation of soil moisture based on the signal strength. In this section, we provide proof-of-concept laboratory measurements of the new antenna design tuned and matched in dry sand, and then placed into the sand with different moisture content related to the summer period results in Figure 6.

If we assume a homogeneous ground, the impedance of dipole buried in the ground can be derived as in [21]. As the measurement results showed good agreement in [22], the sinusoidal current distribution is assumed, and the complex wave number of the soil $k_{soil}$ = $\beta_{soil}$ +i$\alpha_{soil}$, where $\beta_{soil}$ indicates the phase constant and $\alpha_{soil}$ the propagation loss. We have
(3)$$Z_{a}\approx f_{1}\left(\beta_{soil}l\right)-i\left(120\left(ln(2l/d)-1\right)cot\left(\beta_{soil}l\right)-f_{2}\left(\beta_{soil}l\right)\right)$$
for
(4)$$\begin{array}{cc} f_{1}\left(\beta_{soil}l\right)= & -0.4787+7.3246\left(\beta_{soil}l\right)+0.3963{\left(\beta_{soil}l\right)}^{2}+15.6131{\left(\beta_{soil}l\right)}^{3}\\ f_{2}\left(\beta_{soil}l\right)= & -0.4456+17.0082\left(\beta_{soil}l\right)-8.6793{\left(\beta_{soil}l\right)}^{2}+9.6031{\left(\beta_{soil}l\right)}^{3}, \end{array}$$
where *d* is the diameter of the dipole underground, *l* is half of the length of the dipole, and $\beta_{soil}$ is given as
(5)$$\beta_{soil}l=2l\pi /\lambda_{0}Re\left(\sqrt{\epsilon_{soil}}\right)$$
for the relative permittivity of soil $\epsilon_{soil}$ and the wavelength in free space $\lambda_{0}$. In actual IoUT systems, sensors are buried very close to the soil surface (usually 0.15–1 m), so we can expect soil–air interface effects. The theoretical model takes this effect into account, and the impedance of the dipole buried in the subsurface area is given as [1]
(6)$$Z_{aug}=Z_{a}{\left(I_{0}/I_{r}\right)}^{2}$$
as well as the return loss (RL) in dB,
(7)$$RL_{dB}=20log_{10}|\left(Z_{0}+Z_{aug}\right)/\left(Z_{0}-Z_{aug}\right)|,$$
where $I_{0}$ is the current induced on the wire due to the excitation, $I_{r}$ the induced current on the UG dipole (derived in [1]) and $Z_{0}$ is the impedance of the source.

From these expressions, it is clear that the antenna impedance will change when placed underground, the resonant frequency will be shifted according to the change in the wavelength of the soil. Additionally, the permittivity and conductivity of soil are frequency-dependent and time varying in the actual soil types. The losses in the soil, such as losses due to high moisture, will degrade the antenna Q factor and attenuate the signal depending on the degree of water content. It is important to note that when the soil is irrigated, both parameters change the permittivity and conductivity of the soil.

In order to examine these effects further, first we model the antenna in electromagnetic simulation software FEKO (Altair) in free space [23], and then we place the antenna underground in halfspace with sand properties [24]. Similar to the principle used in [25,26], the PIFA is designed as an acronym for "beacon" with letters B, C and N. It achieves omnidirectional pattern, with a gain of 1.5 dBi in free space. The balance between inductance and capacitance of the antenna, i.e., the short circuited part of letter B and the open circuit part (with the rest of the letters whose length dictates the antiresonance frequency and should equal a quarter of a wavelength) is achieved and the antenna radiation pattern and input impedance in free space are given in Figure 7 and Figure 8, respectively. These antennas are highly reactive with narrow fractional bandwidth when tuned in free space, but in lossy medium, the Q factor deteriorates and the antenna fractional bandwidth increases accordingly. The ground plane of the antenna is smaller than the necessary λ/4 in order to fit on small electronic circuits (like LoRa module or Arduino), causing somewhat lower radiation efficiency and bandwidth. In the spectrum of the short-circuited type of antennas (like loops, etc.) the antiresonance appears before resonance, and it is convenient to tune and match the antenna impedance before the antiresonance frequency at 50-ohm point of the antenna resistance frequency characteristic. Moreover, as the antenna impedance is inductive in this part, only the trimmer capacitor in the series is required to match the antenna at the desired frequency (here at 868 MHz).

Figure 7 shows the simulation model of the antenna in free space, printed on FR4 substrate having $\epsilon_{r}=4.4$ and tanδ = 0.02. This antenna has to be reduced because it is placed underground, specifically in dry sand with properties $\epsilon_{r}=4$ and tanδ = 0 [24], causing the wavelength to decrease ($\lambda =c/f(\sqrt{\epsilon_{soil}})$) and the impedance characteristic to shift to lower frequencies. Considering dry sand electrical properties and the aimed antenna resistance of 50 Ω, using (Equation 3)–(Equation 5), it turns out that the length of the equivalent dipole buried in the sand should be 0.68 of the length of the dipole in free space. Finally, after additional parametric study, the underground antenna total size is reduced by scale factor of 0.61. The dimensions of the scaled antenna are very small and close to the electrically small antenna limit, 7.4 cm × 3 cm, in order to fit the electronics of the LoRa module for a small IoUT sensor.

Both impedance characteristics are shown in Figure 8. At 868 MHz, an antenna in free space achieves (49.7 + j312.0) Ω, and a scaled antenna in dry sand obtains Z = (50.3 + j171.6) Ω.

Furthermore, it can be seen that the scaled antenna underground has a lower Q factor (the impedance characteristic is not as sharp as the one in free space) and in the actual soil we can expect this effect, especially if it is a lossy soil (with certain level of moisture). In the next step, the scaled antenna is tuned with the capacitor in series of 1.1 pF, and placed in dry sand and sand with 20% of water content (having $\epsilon_{r}=19$ and tanδ = 0.21 according to [24]). The results in Table 3 show that scaled antenna matched in dry sand achieves significantly less reflection, approximately 40 dB, than in the case when moisture content is 20%. This effect offers a possibility to detect when the soil is irrigated and when it is not (if the water content is below 20% it is considered that it needs to be irrigated, and if it is higher, it does not require irrigation [6]).

## 5. Proof-of-Concept Measurements of the Proposed Antenna

The proposed antenna is fabricated, printed on FR4 substrate and scaled to optimize a match at 868 MHz in sand (see Figure 9 (left)). The trimmer capacitor of 0.8–10 pF in series is used and the capacitance is changed manually by measuring the antenna resonant characteristics with vector network analyzer (VNA). Additionally, the antenna had to be placed in waterproof casing, which also influenced antenna characteristic slightly. The measurement setup, shown in Figure 10 and Figure 11, consisted of a VNA Master MS2028C, VSG25A Vector Signal Generator, BB60C — 6 GHz Real-time Spectrum Analyzer, dipole antenna, SMA cables, and calibration kit, and a plastic container filled with 10 kg of sand. These are indoor laboratory measurements, and onsite measurements of the prototype (Figure 9 (right)) buried in actual soil for a whole year are planned for future work. The focus in our paper is not on the dielectric properties of soil, but rather on the difference between the soil before irrigation in relation to irrigated soil, so the measurements of dielectric constant in sand were not performed. In addition, this type of sand is not dried additionally, and probably has a certain small amount of water content. Water is then added in the sand step by step, 0.5 l, 1 l and 2 l, corresponding to 5, 10, and 20% of water content, respectively. Measurement results in Figure 12. show $\left|S_{11}\right|$ (dB) for different levels of added water content, 0% accounts for the actual sand before irrigation, 5% and 10% for small amounts of water like light rain or water spill, and 20% for irrigated sand. First, the scaled antenna in the sand achieves $\left|S_{11}\right|$ = −14.7 dB at 868 MHz, which can be even better matched if manufacturing losses were minimized. Low water content in the sand shifts the matched frequency right and left but a good match can still be obtained. It is very convenient for this application because signal strength drop should be detected at higher levels of soil moisture. Finally, when the sand reaches 20% of water content, a great difference in $\left|S_{11}\right|$ (RL (dB)) is noted (cca 10 dB) in relation to the case before irrigation. In relation to the RSSI results obtained in summer period with helical LoRa antenna in Figure 6, and based on our previous work in [6], the optimized antenna gives higher signal level differences between the dry and irrigated soil (cca 10 dB but it should be even higher when manufacturing loss is minimized), and consequently should provide higher precision of the moisture-sensing IoT system.

This is also shown in Table 4 where the received power results are given for the buried antenna as a transmitter of 0 dBm at 868 MHz and 1.5 m distance. The measured power at the receiver drops significantly as the content of moisture increases in the sand by 20%. The antenna mismatch in this case is caused by higher loss due to the increased conductivity, and by higher relative permittivity of the soil, as in [12] where the magnitude of electric field at the LoRa gateway position was reduced when moisture content was increased. This difference in signal strength can ensure better detection of soil water content in the UG–AG communication sensor link.

## 6. Conclusions and Future Work

A LoRa-based soil moisture sensing device is used to estimate soil moisture by using the collected signal strength data over a long period of time where the use of an LSTM neural network as a DL approach provided good results in terms of accuracy of soil moisture estimation. As the IoUT systems depend greatly on soil characteristics, the antenna design must be optimized for underground use. The obvious choice for IoUT communication links is usually the wideband antenna design, but here, instead, we propose the narrow-band printed antenna design for the application of soil moisture sensing in order to achieve even better system performance.

By using this approach, all that is required is to conjugate-match the antenna by the capacitor in series in the actual soil before irrigation (or in dry conditions) and obtain a significant signal strength drop caused by mismatch when the soil is irrigated. This can ensure a more reliable radio sensing of soil moisture by using novel sensorless LPWAN technology and various IoUT sensors for agriculture applications.

In future work, the focus will be on considering the possibility of automatically tuning the antenna underground, considering different soil types and onsite measurements with the proposed antenna during a long period of time. Additionally, a possibility of collecting and exploiting the tuning capacitance data in order to increase the system functionality further will also be incorporated in the future investigation.

## Figures and Tables

**Figure 1 sensors-22-09863-f001:**
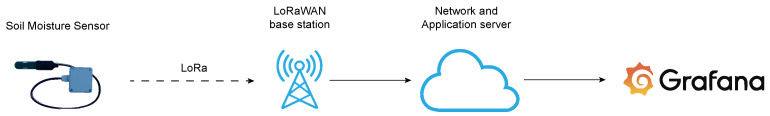
Network architecture of LoRaWAN-based soil moisture sensor system.

**Figure 2 sensors-22-09863-f002:**
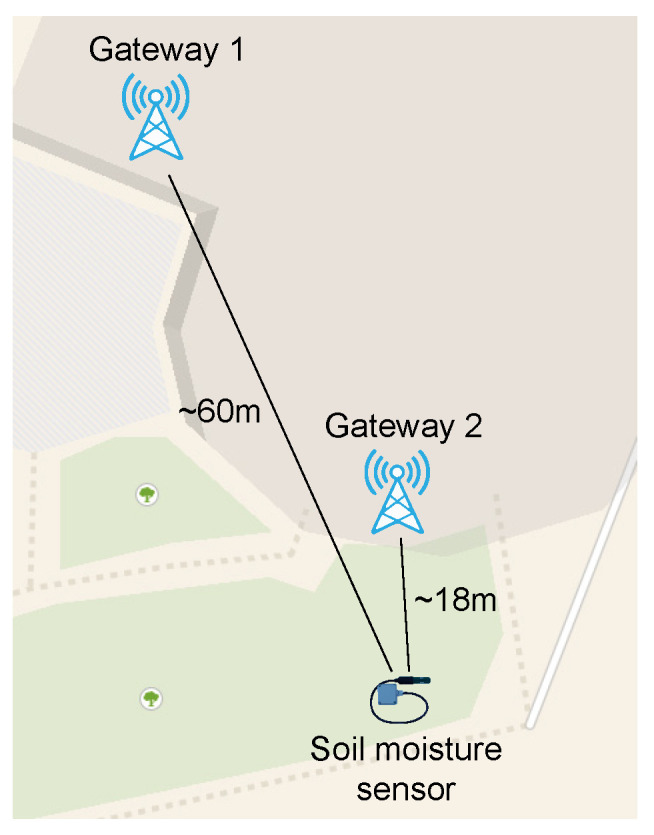
Implementation of LoRaWAN-based soil moisture sensing device.

**Figure 3 sensors-22-09863-f003:**
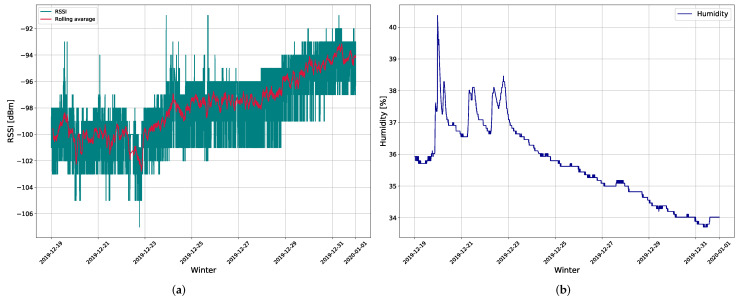
The measurement results of: (**a**) RSSI in dBm collected from gateway 2 and (**b**) soil moisture during the winter period.

**Figure 4 sensors-22-09863-f004:**
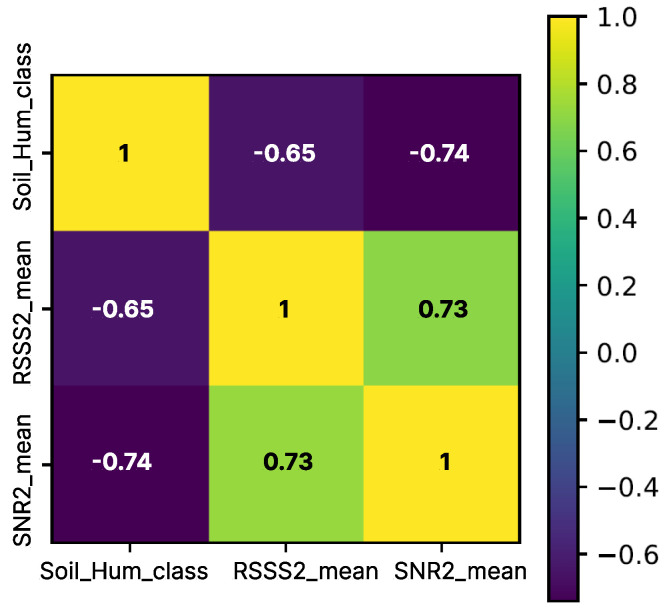
Correlation matrix.

**Figure 5 sensors-22-09863-f005:**
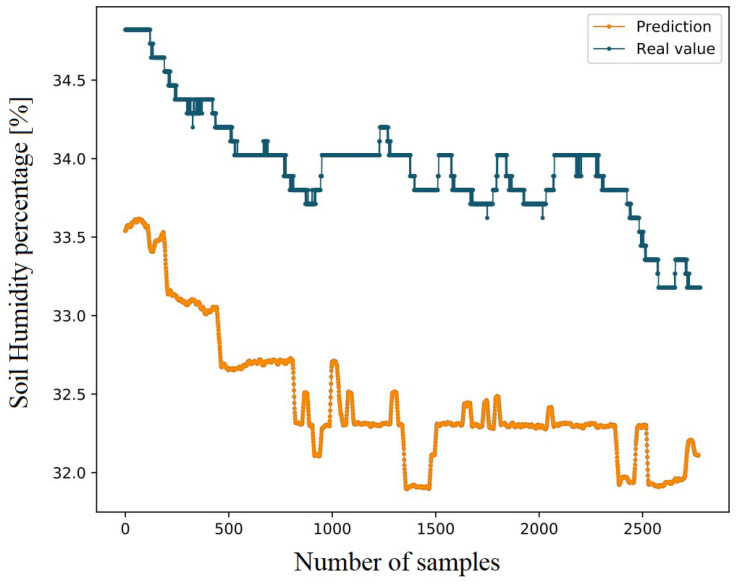
Soil moisture estimation using the SVR model on the test set compared to expected soil moisture values [6].

**Figure 6 sensors-22-09863-f006:**
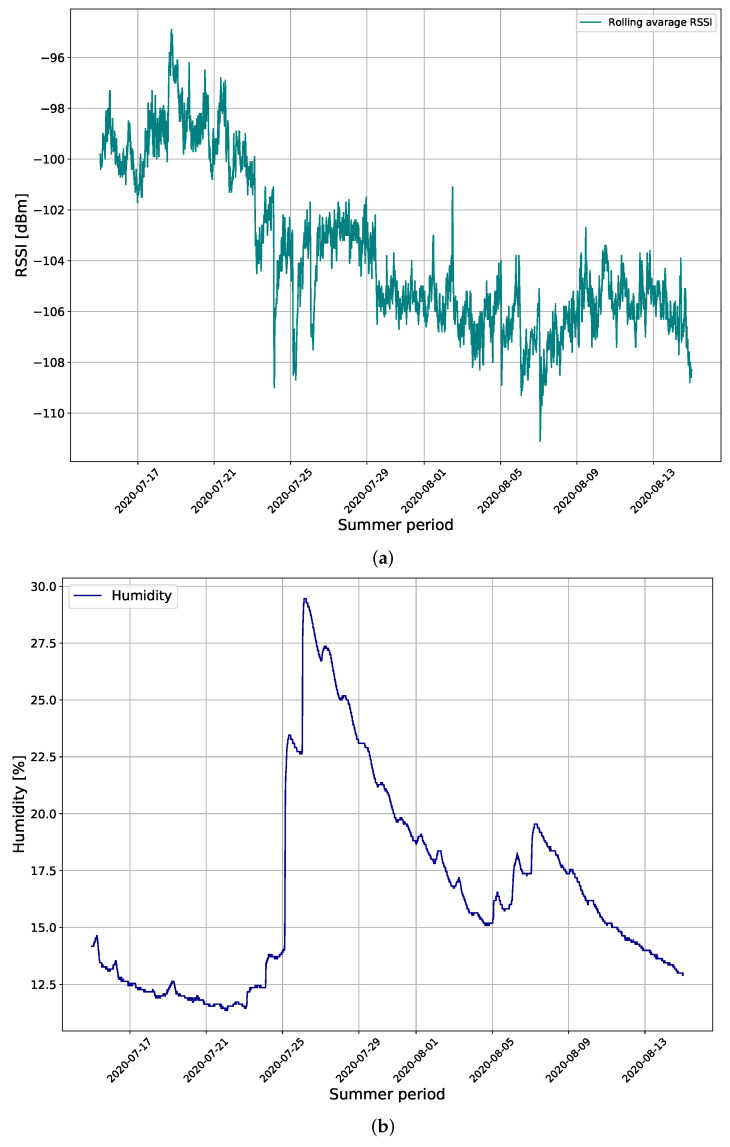
The measurement results of: (**a**) RSSI in dBm collected from gateway 2 and (**b**) soil moisture during the summer period.

**Figure 7 sensors-22-09863-f007:**
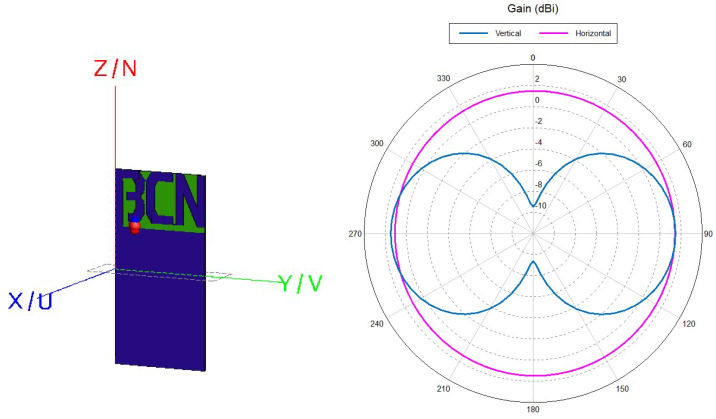
Simulation model of PIFA antenna in FEKO and its 2D radiation pattern for gain in dBi at 868 MHz.

**Figure 8 sensors-22-09863-f008:**
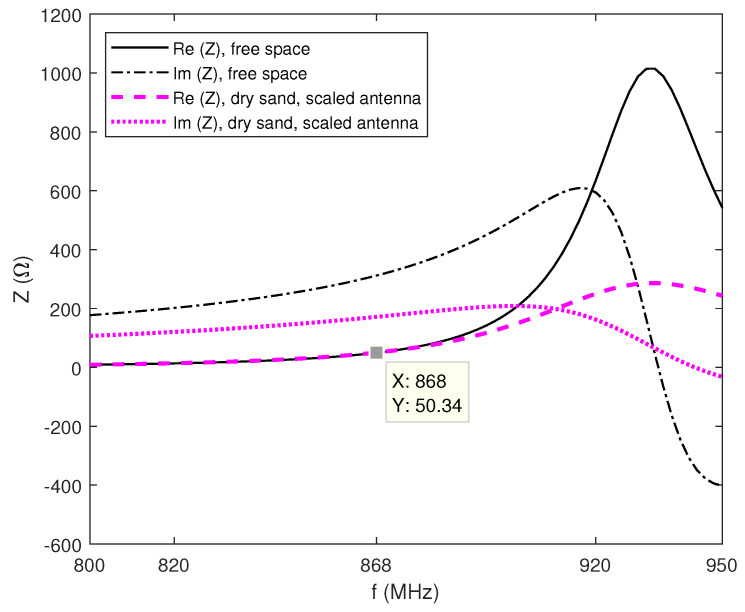
Simulation results for antenna impedance in free space and scaled antenna in dry sand.

**Figure 9 sensors-22-09863-f009:**
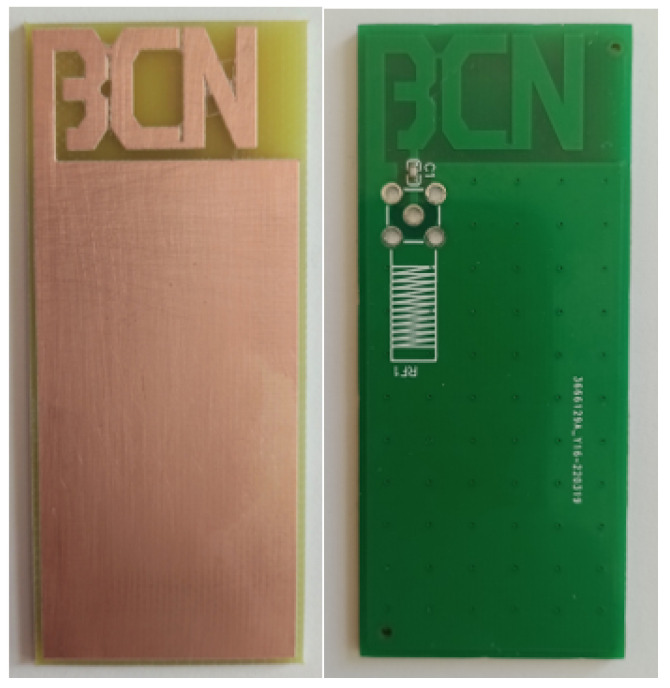
Fabricated PIFA antenna used in laboratory experiments (**left**) and prepared for the prototype testing (**right**).

**Figure 10 sensors-22-09863-f010:**
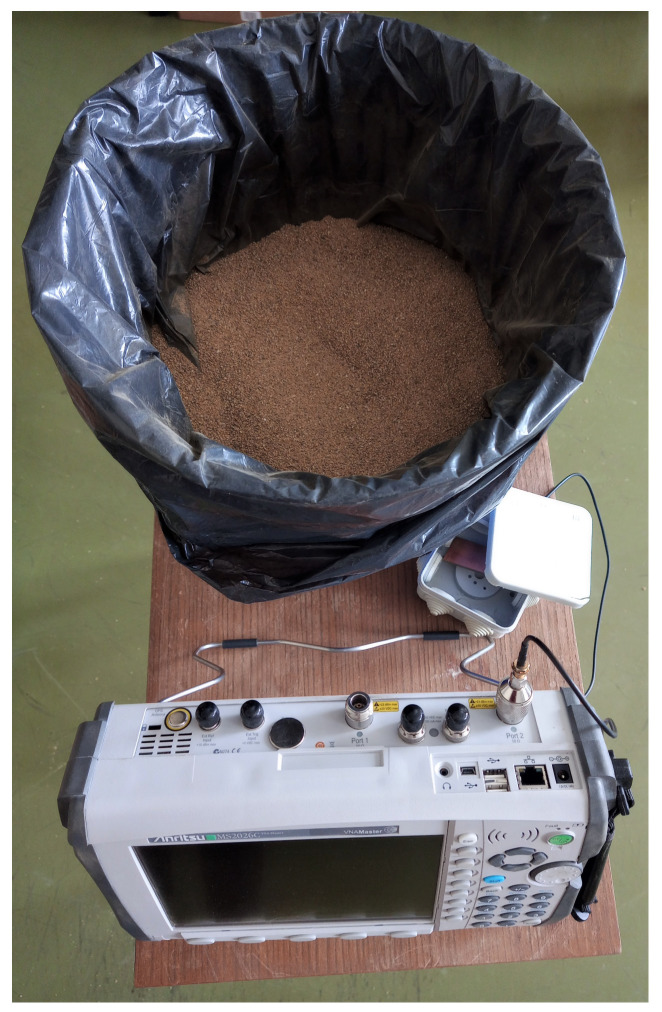
Setup for reflection measurements of antenna characteristics buried in sand.

**Figure 11 sensors-22-09863-f011:**
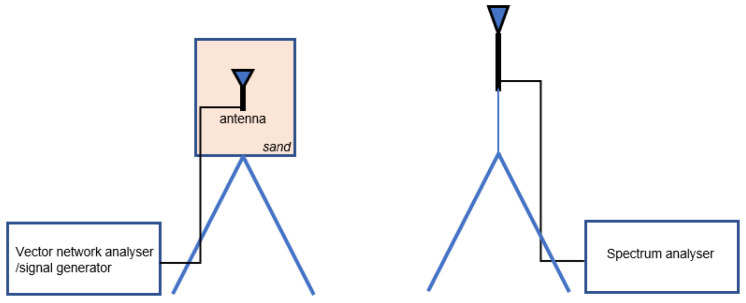
Setup for received signal power measurement.

**Figure 12 sensors-22-09863-f012:**
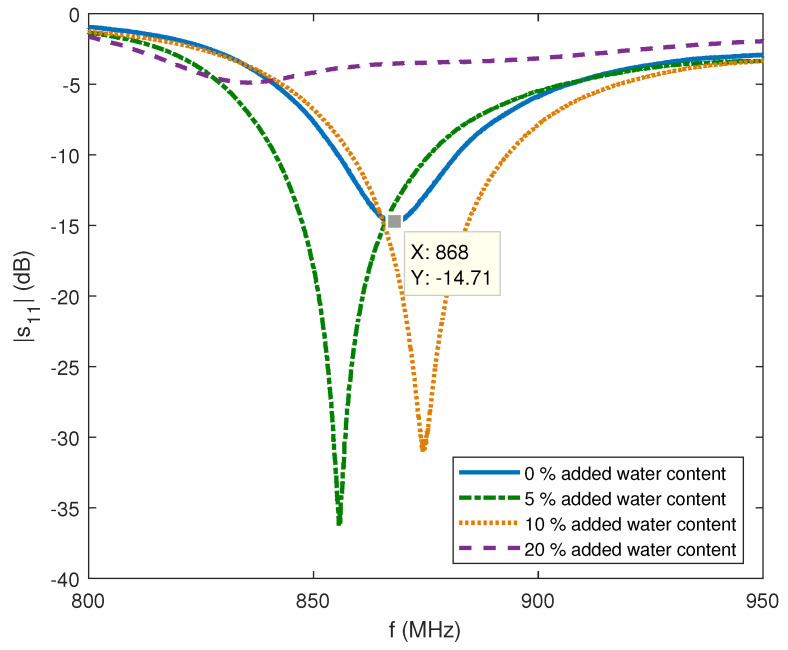
Reflection coefficient measurements of antenna buried in sand with four levels of moisture content.

**Table 1 sensors-22-09863-t001:** Pearson correlation coefficient between soil moisture and RSSI_1_ and SNR_1_.

	RSSI${}_{1}$	SNR${}_{1}$
Soil moisture	−0.29	−0.81

**Table 2 sensors-22-09863-t002:** Comparison parameters of ML algorithms for soil humidity estimation.

Algorithm	Training Time (s)	Test Time (s)	MAE	MSE
SVR	1.451	0.821	0.0487	0.0243
LSTM	1385.992	0.668	0.0104	0.00018

**Table 3 sensors-22-09863-t003:** The antenna reflection results at 868 MHz when buried underground (sand) with varying moisture content.

Moisture Content in Sand	Antenna Impedance (Ω)	RL (dB)
0% added water content	50.3 + j0.2	−47.4
20% added water content	12.2 − j142.0	−0.5

**Table 4 sensors-22-09863-t004:** The received power results at 1.5 m distance and transmitter power of 0 dBm at 868 MHz from the antenna buried in sand with different moisture content.

Moisture Content in Sand	Received Power (dBm)
0 % added water content	−40.96
5 % added water content	−42.82
10 % added water content	−41.89
20 % added water content	−50.22

## Data Availability

Authors can make data available on request. In this case, contact the corresponding author: zblaz@fesb.hr.

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
