# Peer review of "Self-Sensing Antenna for Soil Moisture: Beacon Approach"

_sensors, 2022, doi:10.3390/s22249863_

Round 1

Reviewer 1 Report

The paper presents the self-sensing antenna for soil moisture.

1. This paper reports more about the data processing for soil moisture instead of antenna design and performance (s-parameter and radiation properties).

2. The soil moisture sensing system is proposed with the spring antenna (2nd paragraph of page 3). Why don't the authors use the BCN antenna that is presented in section 3? That should be better if the authors compared the COST spring antenna and the BCN antenna to confirm the soil moisture sensing improvements proposed in the 1st paragraph of section 3.

3. The reviewer proposed to redo the paper structure in order of: 1. Introduction, 2. Hardware design (electronics and also the antenna part), 3 Sensing algorithm, 4. Conclusion.

4. In the soil moisture sensing section, the authors proposed several algorithms, and a comparison table should be reported to make a summary to confirm that the proposal is better. The comparison parameters in the table could be the hardware resource, the estimation time, and the accuracy...

5. In figures 9 and 12, the marker should be added in the curves or change the curve style to make the differences (in the black and white version).

6. In section 3, a series capacitor is used to tune the antenna. How has the capacitance changed?

7. The proposed antenna has a vertical radiation pattern. The paper could be more interesting if the authors made a study on the antenna orientation influence on the system performance.

Author Response

Thank you very much for your valuable comments. Please find all the answers to reviewer's comments in the attached file. 

Reviewer 2 Report

The authors of this article have designed an antenna for soil moisture sensing. This paper is well written. I have a few minor comments.

1. Include quantitative analysis in the abstract of the article.

2. Stress more on novelty in the last paragraph of the introduction section.

3. How antenna dimensions are selected? Are there any equations used or its just based on parametric studies?

4. Include radiation patterns of the antenna with different soil moistures.

Author Response

Thank you very much for your valuable comments. Please find all the answers in the attached file.

Round 2

Reviewer 1 Report

I confirmed the revised version reaches my reviews